# Child acute illness presentation and referrals at primary health clinics in Malawi: a secondary analysis of ASPIRE

Pui-Ying Iroh Tam [1,2] Hussein H Twabi [1] Mtisunge Gondwe,[1] Thomasena O'Byrne,[1] Norman Lufesi,[3] Nicola Desmond [4]

¹Malawi-Liverpool-Wellcome Trust Research Programme, Blantyre, Malawi
²Clinical Sciences, Liverpool School of Tropical Medicine, Liverpool, UK
³Acute Respiratory Illness Unit, Government of Malawi Ministry of Health, Lilongwe, Malawi
⁴International Public Health, Liverpool School of Tropical Medicine, Liverpool, UK

**Correspondence to**
Dr Pui-Ying Iroh Tam;
irohtam@mlw.mw

## ABSTRACT

**Objectives** We aimed to assess the prevalence, presentation and referral patterns of children with acute illness attending primary health centres (PHCs) in a low-resource setting.

**Design, setting and participants** We conducted a secondary analysis of ASPIRE. Children presenting at eight PHCs in urban Blantyre district in southern Malawi with both recorded clinician and mHealth (non-clinician) triage data were included, and patient records from different data collection points along the patient healthcare seeking pathway were consolidated and analysed.

**Results** Between April 2017 and September 2018, a total of 204 924 children were triaged, of whom 155 931 had both recorded clinician and mHealth triage data. The most common presenting symptoms at PHCs were fever (0.3%), cough (0.2%) and difficulty breathing (0.2%). The most common signs associated with referral for under-5 children were trauma (26.7%) and temperature (7.4%). The proportion of emergency and priority clinician triage were highest among young infants <2 months (0.2% and 81.4%, respectively). Of the 3004 referrals (1.9%), 1644 successfully reached the referral facility (54.7%). Additionally, 372 children were sent home from PHC who subsequently self-referred to the referral facility (18.7%).

**Conclusions** Fever and respiratory symptoms were the most common presenting symptoms, and trauma was the most common reason for referral. Rates of referral were low, and of successful referral were moderate. Self-referrals constituted a substantial proportion of attendance at the referral facility. Reducing gaps in care and addressing dropouts as well as self-referrals along the referral pathway could improve child health outcomes.

## INTRODUCTION

The leading global cause of mortality in children under 5 years is preventable infections, namely pneumonia and diarrhoea,[1] and WHO developed the Integrated Management of Childhood Illness (IMCI)[2] to reduce mortality due to these causes. The guidelines provided by IMCI, the integrated Community Case Management (iCCM)[3] and Emergency Triage Assessment and Treatment (ETAT)[4] are part of a comprehensive effort to improve the quality of care of ill children

## STRENGTHS AND LIMITATIONS OF THIS STUDY

⇒ This study prospectively and systematically quantifies child acute illness presentation and referral patterns in an urban district setting in sub-Saharan Africa.
⇒ Analyses a mHealth triage and surveillance dataset of 155 931 children.
⇒ Leverages a mHealth patient surveillance system that tracked those successfully referred to the referral facility, as well as patients that presented to primary health centres, were sent home and then later self-referred to the referral facility.
⇒ Limitations were those inherent in a mHealth surveillance system reliant on existing healthcare workers rather than research-trained staff, including issues of data completeness.

at different levels of the health system in low-resource settings, and to promote access and early initiation of the first-line treatment or referral of severe cases to inpatient facilities, and to reduce mortality.[5–7]

Referral is a critical component of the IMCI strategy, intended to recognise children with severe conditions and ensure appropriate management and optimise quality care at the proper level of the health system.[8 9] In Malawi, in 2013–2014, only 3.5% of hospitals were equipped to provide basic paediatric emergency care in a national facility census.[10] In low-resource settings in sub-Saharan Africa prior to the implementation of IMCI, <2% of children presenting at primary level facilities were referred[8 11]; after the introduction of IMCI, referral rates for children were 6%–20%.[12] Challenges with the referral pathway range from parental education[9 13] to communication,[9] transport,[9 14–16] and cost.[14 16] Therefore, for children who have been referred, successful referral (ie, arrival at the referral facility) has been documented to be as low as 28%.[16]

While social factors affect presentation at and referrals from primary health centres

(PHCs), the clinical features, healthcare utilisation and referral patterns in this setting have been poorly quantified. Elucidating these features can help identify where to allocate scarce resources, and reduce inequities in access. We sought to determine the most common features of acute illness among children presenting to PHCs in an urban, low-resource setting within reasonable proximity to a referral facility, and what features are related to referral and successful referral by age group.

## METHODS

### Study design, setting and participants

The current study is a secondary analysis of data from the Achieving Sustainable Primary Improvement and Engagement in health (ASPIRE) project. At the primary level in Malawi, clinical care is provided by clinically trained health workers (clinicians, nurses, medical assistants) and non-clinically trained health workers (health surveillance assistants, security guards and other health centre staff). Human resource shortages mean that non-clinically trained health workers are often required to fill the gaps in service provision.

ASPIRE implemented a mobile health (mHealth) triage algorithm based on Emergency Triage and Treatment (ETAT) for primary-level care. For ASPIRE, non-clinical health workers positioned at both PHCs and the referral facility (Queen Elizabeth Central Hospital – QECH) were trained and used the mHealth algorithm to document the characteristics of all patients up to 14 years of age, including their symptoms, investigations, triage outcomes and referral statuses as they presented at these sites in urban Blantyre. As part of the same PHC visit, data were first collected by the non-clinical health worker using the mHealth application (mHealth triage), and then separately by the clinical health worker (clinician triage).

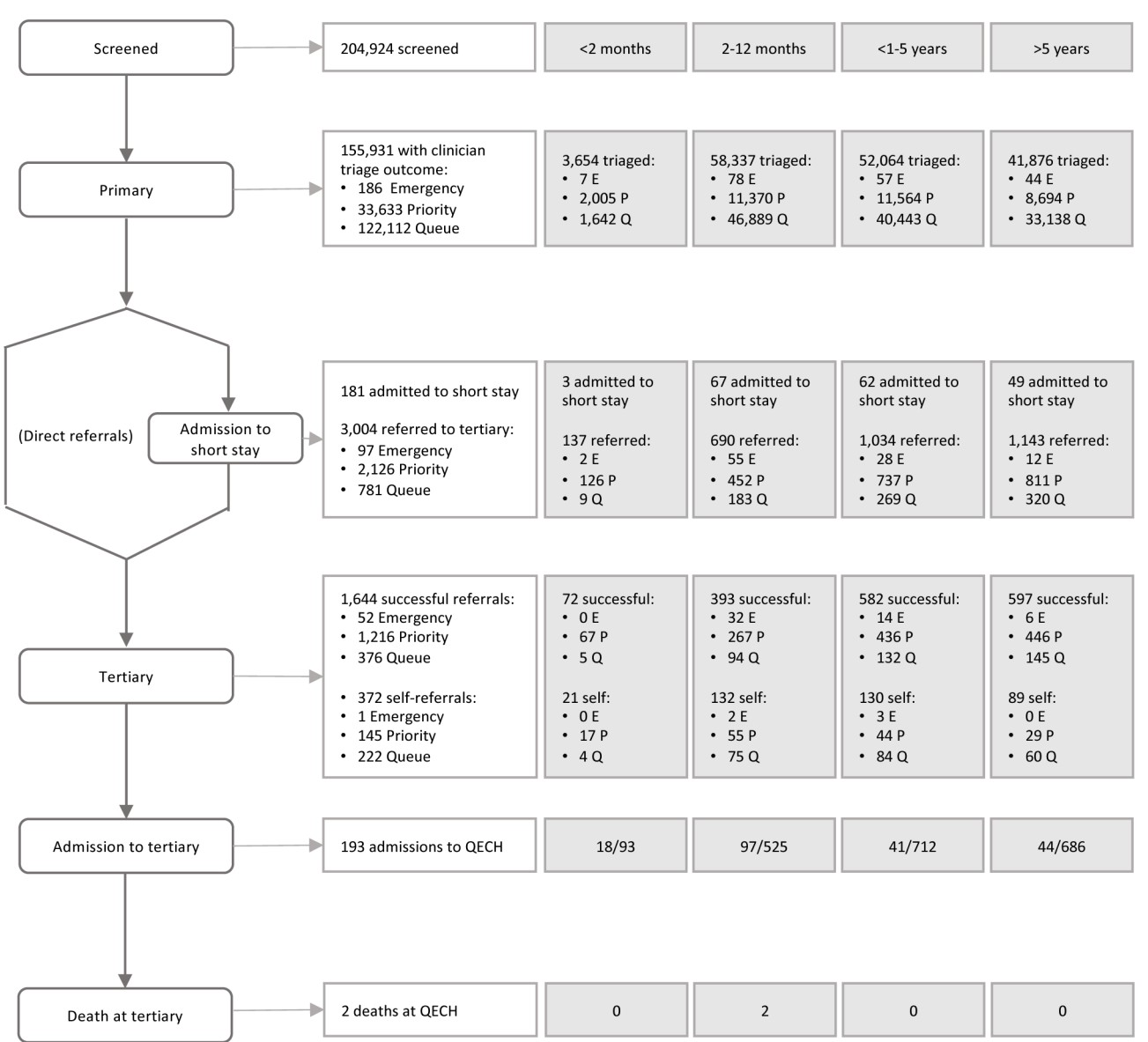

**Figure 1** ASPIRE flow diagram of triage and referral from primary health centre to tertiary facility, by age group.

**Table 1** Main characteristics of children attending primary health centre

| Characteristics | Total n=155931 | % | <2 months n=3654 | % | 2–12 months n=58337 | % | >1–5 years n=52064 | % | >5 years n=41876 | % |
|---|---|---|---|---|---|---|---|---|---|---|
| Male | 77915 | 50.0 | 1889 | 51.7 | 29885 | 51.2 | 26204 | 50.3 | 19937 | 47.6 |
| Weight, kg (mean \| SD) | 15.34 | 8.33 | 4.09 | 1.22 | 8.48 | 2.39 | 13.74 | 3.42 | 24.59 | 7.18 |
| PHC mHealth triage | | | | | | | | | | |
| Emergency | 392 | 0.3 | 19 | 0.5 | 162 | 0.3 | 136 | 0.3 | 75 | 0.2 |
| Priority | 48248 | 30.9 | 2974 | 81.4 | 15694 | 26.9 | 16912 | 32.5 | 12668 | 30.3 |
| Queue | 107291 | 68.8 | 661 | 18.1 | 42481 | 72.8 | 35016 | 67.3 | 29133 | 69.6 |
| PHC clinician triage | | | | | | | | | | |
| Emergency | 186 | 0.1 | 7 | 0.2 | 78 | 0.1 | 57 | 0.1 | 44 | 0.1 |
| Priority | 33633 | 21.6 | 2005 | 54.9 | 11370 | 19.5 | 11564 | 22.2 | 8694 | 20.8 |
| Queue | 122112 | 78.3 | 1642 | 44.9 | 46889 | 80.4 | 40443 | 77.7 | 33138 | 79.1 |
| PHC symptoms | | | | | | | | | | |
| Fever | 515 | 0.3 | 23 | 0.7 | 231 | 0.4 | 166 | 0.3 | 93 | 0.2 |
| Convulsions | 100 | 0.1 | 1 | <0.1 | 26 | <0.1 | 46 | 0.1 | 27 | 0.1 |
| Diarrhoea | 149 | 0.1 | 1 | <0.1 | 86 | 0.1 | 37 | 0.1 | 25 | 0.1 |
| Vomiting | 271 | 0.2 | 9 | 0.2 | 127 | 0.2 | 85 | 0.2 | 50 | 0.1 |
| Cough | 340 | 0.2 | 24 | 0.7 | 201 | 0.3 | 74 | 0.1 | 41 | 0.1 |
| Difficulty breathing | 309 | 0.2 | 26 | 0.7 | 199 | 0.3 | 65 | 0.1 | 19 | <0.1 |
| Rash | 102 | 0.1 | 5 | 0.1 | 26 | <0.1 | 35 | 0.1 | 36 | 0.1 |
| Pallor | 37 | <0.1 | 1 | <0.1 | 10 | <0.1 | 19 | <0.1 | 7 | <0.1 |
| Jaundice | 13 | <0.1 | 2 | 0.1 | 3 | <0.1 | 5 | <0.1 | 3 | <0.1 |
| Oedema | 27 | <0.1 | 0 | 0 | 10 | <0.1 | 11 | <0.1 | 6 | <0.1 |
| Urination problems | 13 | <0.1 | 3 | 0.1 | 1 | <0.1 | 6 | <0.1 | 3 | <0.1 |
| Feeding problems | 97 | 0.1 | 5 | 0.1 | 46 | 0.1 | 27 | 0.1 | 19 | <0.1 |
| Other | 1498 | 1.0 | 50 | 1.4 | 227 | 0.4 | 557 | 1.1 | 664 | 1.6 |
| PHC laboratory test | 503 | 0.3 | 10 | 0.3 | 203 | 0.3 | 168 | 0.3 | 122 | 0.3 |
| Haemoglobin | 46 | <0.1 | 1 | <0.1 | 11 | <0.1 | 20 | <0.1 | 14 | <0.1 |
| Glucose | 7 | <0.1 | 0 | 0 | 1 | <0.1 | 4 | <0.1 | 2 | <0.1 |
| MPS | 16 | <0.1 | 0 | 0 | 3 | <0.1 | 9 | <0.1 | 4 | <0.1 |
| MRDT | 483 | 0.3 | 9 | 0.2 | 201 | 0.3 | 159 | 0.3 | 114 | 0.3 |
| Urine | 4 | <0.1 | 0 | 0 | 0 | 0 | 3 | <0.1 | 1 | <0.1 |
| PHC outcome | | | | | | | | | | |
| Admitted to short stay | 181 | 0.1 | 3 | 0.1 | 67 | 0.1 | 62 | 0.1 | 49 | 0.1 |
| Referred after triage | 2487 | 1.6 | 109 | 3.0 | 562 | 1.0 | 846 | 1.6 | 970 | 2.3 |
| Referred after short stay | 517* | 0.3 | 28 | 0.8 | 128 | 0.2 | 188 | 0.4 | 173 | 0.4 |
| Total referred from HC | 3004 | 1.9 | 137 | 3.7 | 690 | 1.2 | 1034 | 2.0 | 1143 | 2.7 |
| Successful referral† | 1644/3004 | 54.7 | 72/137 | 52.6 | 393/690 | 57.0 | 582/1034 | 56.3 | 597/1143 | 52.2 |
| Self-referral‡ | 372 | 0.2 | 21 | 0.6 | 132 | 0.2 | 130 | 0.2 | 89 | 0.2 |
| QECH A&E | | | | | | | | | | |
| Successful referral arrived by ambulance | 53/1644 | 3.2 | 2/137 | 1.5 | 24/690 | 3.5 | 16/1034 | 1.5 | 11/1143 | 1.0 |
| Self-referral arrived by ambulance | 22/372 | 5.9 | 3/21 | 14.3 | 9/132 | 6.8 | 5/130 | 3.8 | 5/89 | 5.6 |
| Total arrived by ambulance | 75/2016 | 3.7 | 5/93 | 5.4 | 33/525 | 6.3 | 21/712 | 4.4 | 16/686 | 2.3 |
| QECH symptoms | | | | | | | | | | |
| Fever | 205/2016 | 10.2 | 20/93 | 21.5 | 111/525 | 21.1 | 39/712 | 5.5 | 35/686 | 5.1 |

**Table 1** Continued

| Characteristics | Total | | <2 months | | 2–12 months | | >1–5 years | | >5 years | |
|---|---|---|---|---|---|---|---|---|---|---|
| | n=155931 | % | n=3654 | % | n=58337 | % | n=52064 | % | n=41876 | % |
| Convulsions | 32/2016 | 1.6 | 1/93 | 1.1 | 7/525 | 1.3 | 20/712 | 2.8 | 4/686 | 0.6 |
| Diarrhoea | 49/2016 | 2.4 | 2/93 | 2.2 | 39/525 | 7.4 | 4/712 | 0.6 | 4/686 | 0.6 |
| Vomiting | 75/2016 | 3.7 | 5/93 | 5.4 | 41/525 | 7.8 | 13/712 | 1.8 | 16/686 | 2.3 |
| Cough | 159/2016 | 7.9 | 15/93 | 16.1 | 94/525 | 17.9 | 28/712 | 3.9 | 22/686 | 2.8 |
| Difficulty breathing | 148/2016 | 7.3 | 15/93 | 16.1 | 89/525 | 17.0 | 25/712 | 3.5 | 19/686 | 3.2 |
| Rash | 7/2016 | 0.3 | 0 | 0 | 4/525 | 0.8 | 3/712 | 0.4 | 0 | 0 |
| Pallor | 25/2016 | 1.2 | 0 | 0 | 6/525 | 1.1 | 8/712 | 1.1 | 11/686 | 1.6 |
| Jaundice | 11/2016 | 0.5 | 7/93 | 7.5 | 0 | 0 | 3/712 | 0.4 | 1/686 | 0.1 |
| Oedema | 16/2016 | 0.8 | 0 | 0 | 7/525 | 1.3 | 3/712 | 0.4 | 6/686 | 0.9 |
| Urination problems | 3/2016 | 0.1 | 1/93 | 1.1 | 0 | 0 | 1/712 | 0.1 | 1/686 | 0.1 |
| Feeding problems | 52/2016 | 2.6 | 8/93 | 8.6 | 27/525 | 5.1 | 6/712 | 0.8 | 11/686 | 1.6 |
| Other | 117/2016 | 5.8 | 10/93 | 10.8 | 42/525 | 8.0 | 26/712 | 3.7 | 39/686 | 5.7 |
| QECH A&E outcome | | | | | | | | | | |
| Died | 2/2290 | 0.1 | 0/119 | 0 | 2/663 | 0.1 | 0/770 | 0 | 0/738 | 0 |

*517 cases that were referred to QECH from PHC were still admitted to short stay at the PHC, where they were again referred to QECH.
†Referred by PHC and arrived at QECH.
‡Not referred by PHC but arrived at QECH.
A&E, accident and emergency; CSF, cerebrospinal fluid; mHealth, mobile health; MPS, malaria parasite screen; MRDT, malaria rapid diagnostic test; PCV, packed cell volume; PHC, primary health centre; QECH, Queen Elizabeth Central Hospital.

In 2014, the district of Blantyre had a population of 1.9 million,[17] about half of whom were under 15 years of age, with the majority residing in peri-urban townships. QECH is a tertiary-level hospital and referral facility in the southern region of the country; patients are thus expected to first be seen in a PHC or district hospital before presentation to QECH.

### Public involvement
Between 2013 and 2017, the ASPIRE mHealth primary ETAT was developed and tested, with the involvement of district and primary health level stakeholders and national policymakers, and then implemented from 2017 to 2018 across eight PHCs in Blantyre district in southern Malawi. Details of the study have been published previously.[18 19]

### Procedures
An mHealth patient surveillance system was established to monitor patients from initial presentation at the PHC through referral to QECH and final outcome. Participants were entered into the surveillance system as they entered the PHC and assigned a unique identification number (ID), and the mHealth triage outcome following implementation of the algorithm would be recorded by the healthcare worker, together with the facility name, sex and age of the patient. After review by a clinician, the clinician triage and consultation outcome (sent home, admitted for short stay or referred) were documented, as well as any laboratory tests that were conducted. For all referrals, patients were tracked through a standard referral stamp to arrival at QECH. At QECH, presenting features, laboratory tests conducted, final diagnosis and outcome (alive, died, absconded) were documented. A referral was defined as any patient referred from the PHC (either clinic or short stay); successful referral was defined as any patient referred from PHC who arrived at QECH; and self-referral was defined as any patient sent home from PHC who subsequently arrived at QECH.

### Statistical analysis
Prior to analysis, we removed records with invalid patient IDs and duplicated patient IDs that could not be matched unambiguously between data collection points. Records from the different data collection points were matched by patient IDs to allow for patient journey monitoring. Data for the different collection points were extracted as csv files from the database, then analysed in RStudio V.446 (2023.03.1) using R V.4.3.0 (2023-04-21). Descriptive statistics were used to describe the data. Continuous data were described using means and SD when normally distributed, and medians and interquartile ranges (IQR) when non-normal. Categorical data were described using proportions.

### RESULTS
Between April 2017 and September 2018, a total of 204924 children attended eight PHCs in urban Blantyre (figure 1 and online supplemental figure 1), and 155931 had complete clinician and mHealth triage outcome

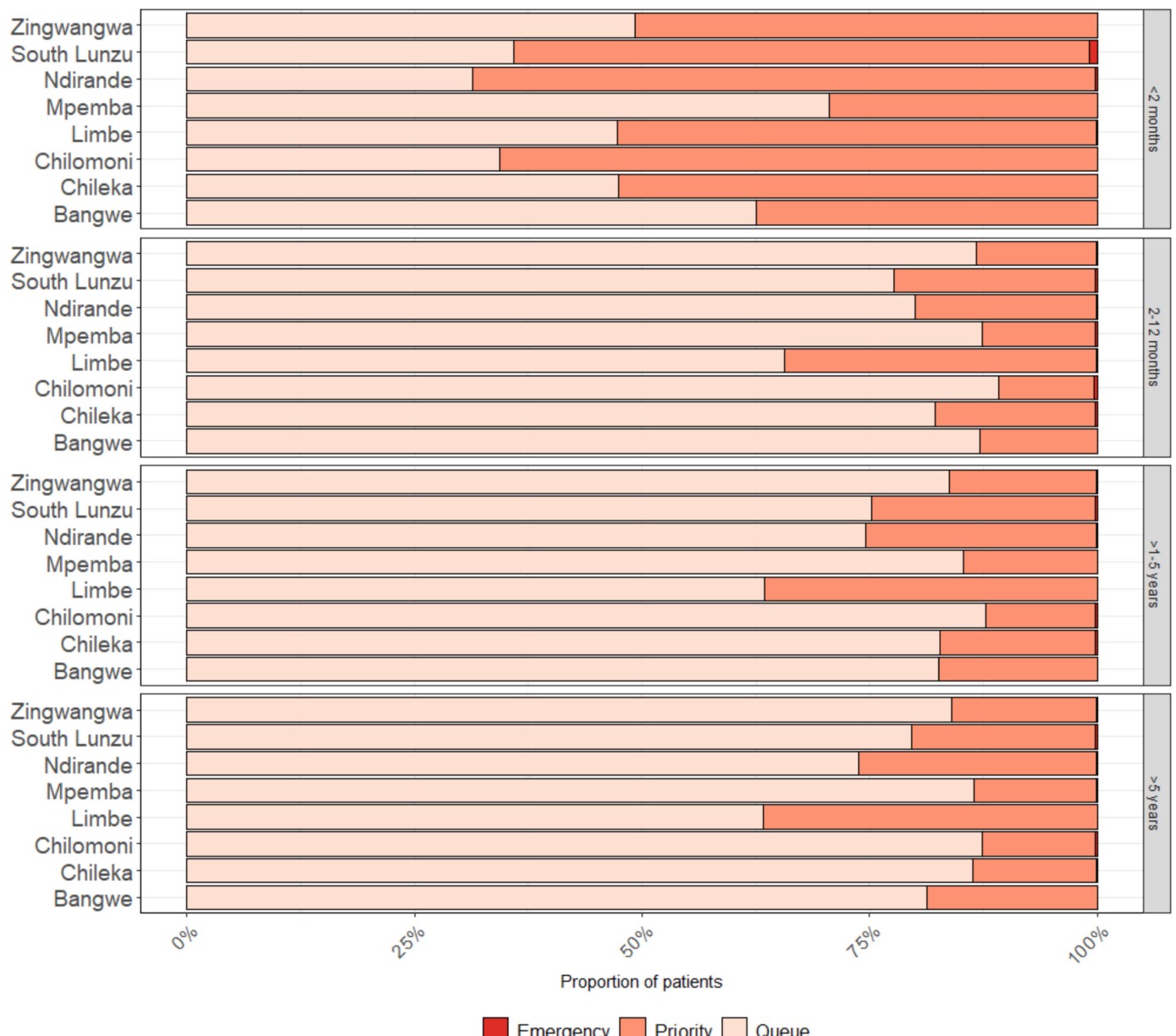

**Figure 2** Proportion triaged as emergency, priority and queue, by age group and by facility.

data (table 1 and online supplemental table 1). Of these, 3654 (2.3%) were <2 months, 58 337 (37.4%) were 2–12 months, 52 064 (33.4%) were >1–5 years and 41 876 (26.9%) were >5 years.

At PHCs, 392 (0.3%) received a mHealth triage of emergency and 48 248 (30.9%) of priority, and 186 received a clinician triage of emergency (0.1%) and 33 633 (21.6%) of priority (figure 2). The proportion of emergency and priority clinician triage were highest among young infants <2 months (7 (0.2%) and 2974 (81.4%) respectively).

At PHCs, the most common documented presenting symptoms were fever (515; 0.3%), cough (340; 0.2%) and difficulty breathing (309; 0.2%). The most commonly performed test was the malaria rapid diagnostic test (MRDT, 483; 0.3%).

There were a total of 3004 referrals (1.9%) to QECH. Within age groups, 3.7% of young infants <2 months presenting were referred, compared with 1.2% of 2–12 months, 2.0% of >1–5 years and 2.7% of >5 years (online supplemental table 2). Infants 2–12 months comprised the largest proportion of emergency triage that were referred (8.0%; online supplemental table 3). Of these referrals, 1644 (54.7%) were successful. For children ≤5 years (table 2), the most common emergency signs related to referral were diarrhoea (130; 7.0%), respiratory distress (35; 1.9%) and obstructed breathing (28; 1.5%), and the most common priority signs were trauma (497; 26.7%), temperature (229; 12.3%) and tiny baby (137; 7.4%); these were also the most common emergency and priority signs related to a successful referral.

At QECH, the most commonly documented diagnoses overall were pneumonia (52/2,016; 2.6%) and gastroenteritis (35/2,016; 1.7%). The most commonly performed tests at QECH were the HIV test (148/2,016;

Table 2 Characteristics of triage outcome and emergency and priority signs with referral for children attending primary health centre for children ≤5 years

| Characteristics | Referred n=1861 | % | Not referred n=112 194 | % | Successfully referred n=1047 | % | Not successfully referred n=815 | % | Self-referred n=283 | % |
|---|---|---|---|---|---|---|---|---|---|---|
| Emergency signs | | | | | | | | | | |
| Breathing | | | | | | | | | | |
| Obstructed breathing | 28 | 1.5 | 102 | 0.1 | 19 | 1.8 | 9 | 1.1 | 0 | 0.0 |
| Central cyanosis | 1 | 0.1 | 4 | <0.1 | 0 | 0 | 1 | 0.1 | 0 | 0 |
| Respiratory distress | 35 | 1.9 | 82 | 0.1 | 21 | 2.0 | 14 | 1.7 | 5 | 1.8 |
| Circulation | | | | | | | | | | |
| Cold hands | 11 | 0.6 | 1214 | 1.1 | 5 | 0.5 | 6 | 0.7 | 7 | 2.5 |
| Capillary refill >3 s | 0 | 0 | 8 | <0.1 | 0 | 0 | 0 | 0 | 0 | 0 |
| Weak and fast pulse | 1 | 0.1 | 20 | <0.1 | 1 | 0.1 | 0 | 0 | 0 | 0 |
| Consciousness | | | | | | | | | | |
| Coma | 3 | 0.2 | 3 | <0.1 | 1 | 0.1 | 2 | 0.2 | 1 | 0.4 |
| Convulsions | 24 | 1.3 | 29 | <0.1 | 15 | 1.4 | 9 | 1.1 | 3 | 1.1 |
| Dehydration | | | | | | | | | | |
| Diarrhoea | 130 | 7.0 | 22 321 | 19.9 | 85 | 8.1 | 45 | 5.5 | 48 | 17.0 |
| Lethargic or unconscious | 6 | 0.3 | 118 | 0.1 | 2 | 0.2 | 4 | 0.5 | 2 | 0.7 |
| Sunken eyes | 16 | 0.9 | 280 | 0.2 | 13 | 1.2 | 3 | 0.4 | 6 | 2.1 |
| Very slow skin pinch | 1 | 0.1 | 60 | 0.1 | 1 | 0.1 | 0 | 0 | 0 | 0.0 |
| Priority signs | | | | | | | | | | |
| Tiny baby | 137 | 7.4 | 3856 | 3.4 | 76 | 7.3 | 62 | 7.6 | 26 | 9.2 |
| Temperature | 229 | 12.3 | 26 390 | 23.5 | 136 | 13.0 | 93 | 11.4 | 74 | 26.1 |
| Trauma | 497 | 26.7 | 1619 | 1.4 | 304 | 29.1 | 193 | 23.7 | 17 | 6.0 |
| Severe pallor | 3 | 0.2 | 241 | 0.2 | 0 | 0 | 3 | 0.4 | 0 | 0 |
| Poisoning | 4 | 0.2 | 19 | <0.1 | 3 | 0.3 | 1 | 0.1 | 0 | 0 |
| Severe pain | 100 | 5.4 | 1050 | 0.9 | 52 | 5.0 | 48 | 5.9 | 14 | 5.0 |
| Respiratory distress | 47 | 2.5 | 798 | 0.7 | 35 | 3.3 | 12 | 1.5 | 9 | 3.2 |
| Restless/lethargic | 5 | 0.3 | 125 | 0.1 | 4 | 0.4 | 1 | 0.1 | 3 | 1.1 |
| Urgent referral | 7 | 0.4 | 16 | <0.1 | 3 | 0.3 | 4 | 0.5 | 0 | 0 |
| Malnutrition | 12 | 0.6 | 41 | <0.1 | 6 | 0.6 | 6 | 0.7 | 2 | 0.7 |
| Oedema | 9 | 0.5 | 46 | <0.1 | 3 | 0.3 | 6 | 0.7 | 0 | 0 |
| Burns | 55 | 0.3 | 274 | 0.2 | 19 | 1.8 | 36 | 4.4 | 2 | 0.7 |
| Triage outcome | | | | | | | | | | |
| Emergency | 85 | 4.6 | 57 | 0.1 | 46 | 4.4 | 39 | 4.8 | 5 | 1.8 |
| Priority | 1315 | 70.7 | 23 624 | 21.1 | 770 | 73.5 | 546 | 67.0 | 116 | 41.0 |
| Queue | 461 | 24.8 | 88 513 | 78.9 | 231 | 22.1 | 230 | 28.2 | 162 | 57.2 |

7.3%), malaria parasite screen and packed cell volume (127/2,016; 6.3% each).

There were 372 self-referrals, the majority of whom were >1–5 years (table 3). Of these, 1 (1.3%) received a clinician triage of emergency, 145 (39.0%) priority and 222 (59.7%) queue. The most common symptoms at PHC among those self-referred were temperature (90; 24.2%) and diarrhoea (54; 14.5%), and the most common diagnoses at QECH were pneumonia (16; 4.3%), malaria (15; 4.0%) and gastroenteritis (14; 3.8%).

Of those successfully referred, 53 arrived by ambulance (3.2%), while 22 who self-referred also arrived by ambulance (5.9%)—this can occur in a setting where an ambulance can provide a dual function to provide local transport in the area.

41 blood and 11 CSF cultures were collected at QECH and 16 were positive (5 blood and 11 CSF). Of the five positive blood cultures, three were triaged as priority and referred, and two triaged as queue, sent home and subsequently self-referred (40%). Of the 11 positive CSF

**Table 3** Characteristics of children discharged from the primary health centre with subsequent self-referral to hospital

| Characteristics | Total n=372 | % | <2 months n=21 | % | 2–12 months n=132 | % | >1–5 years n=130 | % | >5 years n=89 | % |
|---|---|---|---|---|---|---|---|---|---|---|
| Male | 215 | 57.9 | 9 | 42.9 | 72 | 54.5 | 72 | 55.4 | 62 | 69.7 |
| Weight, kg (mean | SD) | 11.80 | 4.18 | – | – | 4.90 | – | 13.03 | 1.95 | 15.00 | – |
| PHC mHealth triage | | | | | | | | | | |
| Emergency | 11 | 3.0 | 0 | 0 | 5 | 3.8 | 5 | 3.8 | 1 | 1.1 |
| Priority | 197 | 53.0 | 21 | 100 | 68 | 51.5 | 58 | 44.6 | 50 | 56.2 |
| Queue | 164 | 44.1 | 0 | 0 | 59 | 44.7 | 67 | 51.5 | 38 | 42.7 |
| PHC clinician triage | | | | | | | | | | |
| Emergency | 1 | 1.3 | 0 | 0 | 2 | 1.5 | 3 | 2.3 | 0 | 0 |
| Priority | 145 | 39.0 | 17 | 81.0 | 55 | 41.7 | 44 | 33.8 | 29 | 32.6 |
| Queue | 222 | 59.7 | 4 | 19.0 | 75 | 56.8 | 83 | 63.8 | 60 | 67.4 |
| Emergency signs | | | | | | | | | | |
| Breathing | | | | | | | | | | |
| Obstructed breathing | 1 | 0.3 | 0 | 0 | 0 | 0 | 0 | 0 | 1 | 1.1 |
| Central cyanosis | 0 | 0 | 0 | 0 | 0 | 0 | 0 | 0 | 0 | 0 |
| Respiratory distress | 5 | 1.3 | 0 | 0 | 2 | 1.5 | 3 | 2.3 | 0 | 0 |
| Circulation | | | | | | | | | | |
| Cold hands | 9 | 2.4 | 0 | 0 | 5 | 3.8 | 2 | 1.5 | 2 | 2.2 |
| Capillary refill >3 s | 0 | 0 | 0 | 0 | 0 | 0 | 0 | 0 | 0 | 0 |
| Weak and fast pulse | 0 | 0 | 0 | 0 | 0 | 0 | 0 | 0 | 0 | 0 |
| Consciousness | | | | | | | | | | |
| Coma | 1 | 0.3 | 0 | 0 | 1 | 0.8 | 0 | 0 | 0 | 0 |
| Convulsions | 3 | 0.8 | 0 | 0 | 2 | 1.5 | 1 | 0.8 | 0 | 0 |
| Dehydration | | | | | | | | | | |
| Diarrhoea | 54 | 14.5 | 3 | 14.3 | 35 | 26.5 | 10 | 7.7 | 6 | 6.7 |
| Lethargic or unconscious | 2 | 0.5 | 0 | 0 | 1 | 0.8 | 1 | 0.8 | 0 | 0 |
| Sunken eyes | 6 | 1.6 | 0 | 0 | 5 | 3.8 | 1 | 0.8 | 0 | 0 |
| Very slow skin pinch | 0 | 0 | 0 | 0 | 0 | 0 | 0 | 0 | 0 | 0 |
| Priority signs | | | | | | | | | | |
| Tiny baby | 26 | 7.0 | 19 | 90.5 | 6 | 4.5 | 1 | 0.8 | 0 | 0 |
| Temperature | 90 | 24.2 | 2 | 9.5 | 40 | 30.3 | 32 | 24.6 | 16 | 18.0 |
| Trauma | 36 | 9.7 | 0 | 0 | 5 | 3.8 | 12 | 9.2 | 19 | 21.3 |
| Severe pallor | 0 | 0 | 0 | 0 | 0 | 0 | 0 | 0 | 0 | 0 |
| Poisoning | 0 | 0 | 0 | 0 | 0 | 0 | 0 | 0 | 0 | 0 |
| Severe pain | 23 | 6.2 | 0 | 0 | 5 | 3.8 | 9 | 6.9 | 9 | 10.1 |
| Respiratory distress | 11 | 3.0 | 0 | 0 | 7 | 5.3 | 2 | 1.5 | 2 | 2.2 |
| Restless/lethargic | 5 | 1.3 | 0 | 0 | 2 | 1.5 | 1 | 0.8 | 2 | 2.2 |
| Urgent referral | 1 | 0.3 | 0 | 0 | 0 | 0 | 0 | 0 | 1 | 1.1 |
| Malnutrition | 2 | 0.5 | 0 | 0 | 2 | 1.5 | 0 | 0 | 0 | 0 |
| Oedema | 0 | 0 | 0 | 0 | 0 | 0 | 0 | 0 | 0 | 0 |
| Burns | 3 | 0.8 | 0 | 0 | 1 | 0.8 | 1 | 0.8 | 1 | 1.1 |
| QECH diagnosis | | | | | | | | | | |
| Trauma | 1 | 0.3 | 0 | 0 | 0 | 0 | 0 | 0 | 1 | 1.1 |
| Gastroenteritis | 14 | 3.8 | 1 | 4.8 | 12 | 9.1 | 1 | 0.8 | 0 | 0 |
| Pneumonia | 16 | 4.3 | 1 | 4.8 | 12 | 9.1 | 3 | 2.3 | 0 | 0 |

Continued

**Table 3**  Continued

| Characteristics | Total | | <2 months | | 2–12 months | | >1–5 years | | >5 years | |
|---|---|---|---|---|---|---|---|---|---|---|
| | n=372 | % | n=21 | % | n=132 | % | n=130 | % | n=89 | % |
| Meningitis | 2 | 0.5 | 0 | 0 | 1 | 0.8 | 1 | 0.8 | 0 | 0 |
| Malnutrition | 10 | 2.7 | 0 | 0 | 6 | 4.5 | 2 | 1.5 | 2 | 2.2 |
| Sepsis | 4 | 1.1 | 3 | 14.3 | 1 | 0.8 | 0 | 0 | 0 | 0 |
| Malaria | 15 | 4.0 | 0 | 0 | 5 | 3.8 | 7 | 5.4 | 3 | 3.4 |
| Anaemia | 6 | 1.6 | 0 | 0 | 3 | 2.3 | 2 | 1.5 | 1 | 1.1 |
| Other | 55 | 14.8 | 4 | 19.0 | 25 | 18.9 | 12 | 9.2 | 14 | 15.7 |
| **QECH laboratory test result** | | | | | | | | | | |
| Positive blood culture | 2 | 0.5 | 0 | 0 | 0 | 0 | 0 | 0 | 2 | 2.2 |
| Positive CSF culture | 6 | 1.6 | 1 | 4.8 | 2 | 1.5 | 2 | 1.5 | 1 | 1.1 |
| Positive HIV test | 0 | 0 | 0 | 0 | 0 | 0 | 0 | 0 | 0 | 0 |
| Positive MRDT | 7 | 1.9 | 0 | 0 | 2 | 1.5 | 3 | 2.3 | 2 | 2.2 |
| Positive MPS | 9 | 2.4 | 0 | 0 | 6 | 4.5 | 1 | 0.8 | 2 | 2.2 |
| Low blood glucose | 2 | 0.5 | 0 | 0 | 1 | 0.8 | 0 | 0 | 1 | 1.1 |
| PCV, (mean | SD) | 32.34 | 6.61 | 36.25 | 1.89 | 31.00 | 5.89 | 35.60 | 2.30 | 30.80 | 11.99 |
| **QECH A&E outcome** | | | | | | | | | | |
| Died | 2 | 0.5 | 0 | 0 | 2 | 1.5 | 0 | 0 | 0 | 0 |

A&E, accident and emergency; CSF, cerebrospinal fluid; mHealth, mobile health; MPS, malaria parasite smear; MRDT, malaria rapid diagnostic test; PCV, packed cell volume; PHC, primary health clinic; QECH, Queen Elizabeth Central Hospital.

cultures, one was triaged as emergency and referred, six triaged as priority and three referred, and five triaged as Queue and two referred; the remainder (45%) were sent home and then self-referred to QECH. Seven of 16 positive cultures (44%) were among children 2 months of age or less.

Two infants 2 months of age died, both at QECH. One was a female that received a clinician triage of queue at the PHC and was sent home, then self-referred to QECH. At QECH, she was noted to have fever, cough, difficulty breathing and 'other' problem. CSF culture, blood culture, HIV, MRDT and other labs were collected, and the child was given a diagnosis of 'other.' The other was a male who received a clinician triage of priority and was sent home, then self-referred to QECH, where he was noted to have oedema and 'other' problem, and was given a diagnosis of 'other.' CSF culture, blood culture, HIV, MRDT, glucose, and 'other' labs were collected. CSF cultures were positive for both; however, as the PHC patient identifiers did not link up with the microbiology database, further information was not retrieved on the cultures.

## DISCUSSION

This study is, to our knowledge, the largest description of child acute illness presentation at primary level clinics and the first to prospectively and comprehensively describe clinical features, and quantify healthcare utilisation and referral patterns in sub-Saharan Africa. We documented over 155 000 children attending PHCs in urban Blantyre, and noted a referral rate of <2%. Though infants 2–12 months had the highest absolute numbers of attendance, young infants <2 months accounted for the largest proportion of emergency and priority signs, and the highest proportion of referrals, with fever and respiratory symptoms being the most common symptoms.

Comprehensive collection of triage at PHCs and prospective tracking of referrals to successful arrival at the referral hospital in a low-resource setting has not been documented previously. Studies of referral patterns in Uganda, Tanzania, Democratic Republic of the Congo and Nigeria have been based either on retrospective review of records, interviews, focus group discussions or follow-up visits, and documented successful referral rates ranging from 0.6% to 67%.[8 16 20] The largest prospective evaluation of successful referral involved prereferral rectal artesunate among 6400 children with suspected severe malaria in the Democratic Republic of the Congo, Nigeria and Uganda, and determined referral completion with a 28-day follow-up visit.

This study is the first to report successful referrals that have been quantified in a systematic way. We used an mHealth patient surveillance system that allowed us not only to track successful referrals to QECH but also to identify patients that had been seen in PHCs, sent home and then self-referred to QECH. Self-referrals made up 18.7% of overall referrals, and to our knowledge is the first time this has been quantified prospectively. Self-referral patterns has been described in other sub-Saharan African settings but those studies have focused

on bypassing PHCs entirely,[21–23] rather than self-referral after being sent home from PHC.

We found that young infants were not the largest proportion of attendees at PHCs, but they did account for the largest proportion of emergency and priority signs, and referrals. This supports findings from other studies of sub-Saharan Africa where, compared with those 2–59 months, a higher proportion of young infants <2 months were referred from first-level health facilities in Uganda, Tanzania and Nigeria.[12] A systematic review of community-to-facility neonatal referral rates in Africa reported high completion rates ranging from 74%–93%.[24] Given that neonatal mortality remains high despite gains made by the Sustainable Development Goals, this highlights the importance of the health system referral pathway to support the care of the young infant age group, and suggests that allocation of resources should target young infants as a vulnerable population.

For a healthcare worker applying IMCI guidelines, IMCI symptom referral criteria have a demonstrated sensitivity of 42%–74% and specificity of 93%–99%.[25] In our study, the most common symptoms at PHC were fever followed by respiratory symptoms, which corresponds with the the most common diagnoses at the referral hospital of pneumonia and malaria. Malaria was investigated for the most frequently at PHCs and diagnosed commonly at tertiary level for children greater than 2 months of age. This reflects the continued importance of infectious diseases on child morbidity and mortality. Trauma accounted for a sizeable proportion of cases, becoming the predominant cause of referral after the first year of life, highlighting the role and burden of injury in child health.[26]

We collected data on triage, since this has been shown to reduce inpatient mortality by up to two-thirds in low-resource settings.[7] Notably, 59.8% (85/142) of emergency, 5.3% (1315/24 939) priority and 0.5% (461/88 974) queue triage cases were referred, which would suggest that there are opportunities to strengthen the IMCI, ETAT and referral training of healthcare workers. Key health system constraints lead to poor quality paediatric care,[27] and prior assessment of inpatient paediatric care in first-referral level hospitals in Kenya showed that case management guidelines are often not in line with national or international guidelines,[28] which may explain why there were lower than expected rates of referral. It is noteworthy that the two deaths that occurred were both in children 2–12 months of age that had been triaged at PHC, one as priority and the other as queue, and then sent home before self-referring to QECH. Given that the Malawi under-5 mortality rate in 2017 was 51 per 1000 live births, the documentation of only two deaths in this surveillance is low, and may reflect the limitations of the surveillance platform used, as well as prior observations that over half of child deaths were observed to occur at home, and that cases reaching the PHC and hospital may represent only the tip of the iceberg.[8]

This study has limitations. As part of the implementation work around the mHealth surveillance system, existing healthcare workers based at PHCs were trained to collect data as part of their routine duties but were not research staff trained at the level to conduct a clinical trial. Therefore, the pragmatic approach resulted in issues of data completeness and did not allow us to interrogate the data more closely. The referral pathway relied on tracking patients through a standard referral stamp; therefore, patients who bypassed the PHC triage system due to acute illness or injury, or who misplaced this stamp would therefore not be captured at the referral hospital. The lack of data completeness may therefore affect the interpretation of results by leading to an underestimation of cases and successful referrals. The mHealth surveillance system was based on patient identifiers that were separate from the QECH system, and therefore we were unable to retrieve the results of investigations to fully chronicle the patient journey and cross-check results.

Nevertheless, this study is one of the first to prospectively and systematically quantify child acute illness presentation, healthcare utilisation and referral patterns in an urban district setting in a low-resource setting. A strong health system providing quality health service requires an effective referral pathway, to promote timely access, judicious allocation of human and physical resources, cost-effective care, and to optimise clinical outcomes. Our study provides quantitative data that allows us to identify gaps in care, which can form the basis to strengthen the referral pathway through regular monitoring, evaluation and feedback to improve child health outcomes.

**Acknowledgements** We thank Marc Henrion for assistance with the ASPIRE dataset and statistical analysis.

**Contributors** PI conceived, led and wrote the first draft of the manuscript. HHT conducted the statistical analysis. MG, TOB, NL and ND led the primary ASPIRE study. All authors reviewed and approved the final manuscript.PI accepts full responsibility for the work and/or the conduct of the study, had access to the data, and controlled the decision to publish

**Funding** Meningitis Research Foundation (CSF 19-17 to ND), the Scottish Government (M/15/H/005 to ND), Irish AID, and Wellcome (206545/Z/17/Z).

**Competing interests** None declared.

**Patient and public involvement** Patients and/or the public were involved in the design, or conduct, or reporting, or dissemination plans of this research. Refer to the Methods section for further details.

**Patient consent for publication** Not applicable.

**Ethics approval** Ethical approval for this study was obtained from the College of Medicine Research Ethics Committee (P·.09/16/2021). Since the only patient identifier collected was age, patient-level consent was waived. Facility-level consent was obtained for data collection through the mHealth tool. All data were analysed at aggregate level and anonymised before analysis.

**Provenance and peer review** Not commissioned; externally peer reviewed.

**Data availability statement** Data are available upon reasonable request. An anonymised, de-identified version of the dataset can be made available on request to allow all results to be reproduced. All requests should be directed to the corresponding author.

terminology, drug names and drug dosages), and is not responsible for any error and/or omissions arising from translation and adaptation or otherwise.

**ORCID iDs**
Pui-Ying Iroh Tam http://orcid.org/0000-0002-3682-8892
Hussein H Twabi http://orcid.org/0000-0003-4473-296X
Nicola Desmond http://orcid.org/0000-0002-2874-8569

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
