## [Reviewer comments · BMJ Open]

ARTICLE DETAILS

TITLE (PROVISIONAL)	Child acute illness presentation and referrals at primary health clinics in Malawi: a secondary analysis of ASPIRE
AUTHORS	Iroh Tam, Pui-Ying; Twabi, Hussein; Gondwe, Mtisunge; O'Byrne, Thomasena; Lufesi, Norman; Desmond, Nicola

VERSION 1 – REVIEW

REVIEWER	Vallersnes, Odd Martin University of Oslo, Department of General Practice
REVIEW RETURNED	31-Oct-2023

GENERAL COMMENTS	Your manuscript presents a study of the referral patterns of children presenting at PHCs in a region of Malawi. It is an impressive study, including a very large number of children. As it has not been previously done, describing the patient flow in this setting is highly commendable, and should form a basis for delivery of care and future research. The prospective design and the ability to track patients from the PHCs to the hospital level are features contributing to the quality of the study. However, there are some issues that need to be addressed. The most important is what seems to be a lack of data completeness, as you briefly mention in your Limitations paragraph. What puzzles me is that very few patients are registered with a symptom at the PHC or the QECH (table 1), or a diagnosis at the QECH (tables 1 & 3). What about all the other patients? Did they not have symptoms? Or were they not diagnosed? Or are the data missing? This needs to be addressed and explained. If the data is missing, I am not sure that you should include all the patients in these analyses. What is the difference between PHC mHealth triage and PHC clinician triage? Were they used for the same or different purposes? Were they done at (more or less) the same time? Why do they yield different outcomes? This needs to be explained. Abstract Methods: Needs rewriting. In its current form it does not give the information needed to understand the how the study was done. Abstract Results: The meaning of the term “successful referral” needs an explanation. In table 1, PHC outcome, the sum of total referred from PHC does not equal the number referred after triage + referred after short stay. This needs an explanation or a correction.
---

	Discussion, p11 line 17-19: the sensitivity and specificity of the IMCI criteria: To what gold standard were they related? Discussion, p11 line 42-45: These are not the same numbers as in table 1/figure 1. Discussion, p11 line 52-57: "case management guidelines are often not in line with national or international guidelines, which may explain our findings." How does this explain your findings. Please elaborate. Discussion, final paragraph: In your concluding remarks you state that "Our study provides quantitative data that allows us to identify gaps in care, which can form the basis to strengthen the referral pathway and improve child health outcomes." Please be more concrete: Which gaps in care? How would you suggest to go about to strengthen the referral pathway? Supplementary tables 1 and 2 were missing.
--	---

REVIEWER	van Houten, Marlies Sparne Gasthuis
REVIEW RETURNED	05-Nov-2023

GENERAL COMMENTS	good things:  - great way to track patients from remote PHCs to hospital in a low resource setting - Great to quantify self-referral percentage - great to see percentage of participants at each triage level referred or not - The reviewer provided a marked copy with additional comments. Please contact the publisher for full details.
--

REVIEWER	O'Regan, Andrew University of Limerick Faculty of Education and Health Sciences, University of Limerick School of Medicine
REVIEW RETURNED	06-Nov-2023

GENERAL COMMENTS	I wish to commend you on an interesting, important and well conducted study. There are a number of points that need to be clarified before it is ready for publication in the BMJ Open. I have included these before and am confident that when they have been incorporated that you will have a strong, readable paper that will be of interest to a wide audience, and will make an impact. While this is an excellent study and should be published, there are a number of important clarifications and revisions required to make the current draft more readable for a wider audience. The title does not flow as well as it could- consider a slight rewording, such as 'paediatric presentations and referrals at a primary health centre in Malawi – a secondary analysis of ASPIRE' Abstract It is not clear to me from the abstract what the context and exact purpose of the study is. There is a
--

	300 word count limit and consider expanding the extract to explain what is meant by a successful referral. The results subsection of the abstract is confusing. If the introduction is re-written to state 'the aim of the study is to investigate the most common presentations, diagnoses and reasons for referral', then ensure that the Results section corresponds to the stated aim. What this study adds Lines 24-31: does the 'referral rate' mean that of all the presentations to the PHCs that <2% were referred on to hospital? If yes, it should be explicitly stated. Does the line "...were sent, home and then self-referred to the referral centre" mean that they were discharged from the PHC and later attended a hospital? The wording is not clear. The term successful referral is explained later in the introduction but it would not be understood as it stands in the abstract. Introduction P 4 Line 6 – "under – 5 deaths" – rephrase please to – "mortality/death in children under five years" P 4 line 56 – similarly, rephrase the term 'under-five children' Overall, I think the introduction could be enhanced by more reference to how children are assessed in community and in-hospital settings in SubSaharan Africa. As it stands, a reader unfamiliar with the subject would not have a strong enough context from the introduction. The authors identify the evidence gap in the literature and make the case for the study but the aim should be clearly stated and it would help if a sentence outlining the aim was followed by three bullet points that articulate clearly what the specific objectives are. Methods P 5 ETAT is used for the first time in the main manuscript – the full term should be used followed by ETAT in brackets (this has been done in the abstract but needs to be done again in the main manuscript). The terminology is confusing for the reader who is not familiar with SubSaharan Africa health care systems. The authors could clarify what is a primary health centre – how is it different to what is called a primary health care centre in Ireland or Britain? Are they specifically catered for children under five, or all children or everyone, are they walk-in, appointment-only, free, and who runs them doctors or trained physician assistants or nurses? Similarly, is the term referral facility the standard term? Would referral hospital be more self-explanatory and do they mean specific paediatric units or general A and E? Results From the introduction section, it appears that the study is concerned with children under five years but the results section appears to include the 41,876 children aged over five years. This needs to be clarified. Discussion P9 line 42 – type 'o' -remove one 'is'.
--	--

	Phraseology needs to be improved in several places to ensure accuracy of what is being stated, e.g., 'This study is the first time that successful referrals have been quantified in a systematic way along the referral pathway'... consider alternatively 'this study is the first to report successful referrals that have been quantified in a systematic way'. That is how I understand it. The limitations section is very good but the following paragraph should be rewritten as a conclusion so it should state exactly what the study contributed, i.e. rather than stating a fact that is already known "an effective referral pathway, to promote timely access, judicious allocation of human and physical resources, cost-effective care, and to optimise clinical outcomes", that you have identified a gap in the pathway or rather a way of quantifying it and that this will contribute to a stronger, more seamless referral pathway.
--	--

VERSION 1 – AUTHOR RESPONSE

Reviewer: 1

Dr. Odd Martin Vallersnes, University of Oslo, Oslo Accident and Emergency Outpatient Clinic

Comments to the Author:

Dear authors

Your manuscript presents a study of the referral patterns of children presenting at PHCs in a region of Malawi. It is an impressive study, including a very large number of children. As it has not been previously done, describing the patient flow in this setting is highly commendable, and should form a basis for delivery of care and future research. The prospective design and the ability to track patients from the PHCs to the hospital level are features contributing to the quality of the study.

However, there are some issues that need to be addressed. The most important is what seems to be a lack of data completeness, as you briefly mention in your Limitations paragraph. What puzzles me is that very few patients are registered with a symptom at the PHC or the QECH (table 1), or a diagnosis at the QECH (tables 1 & 3). What about all the other patients? Did they not have symptoms? Or were they not diagnosed? Or are the data missing? This needs to be addressed and explained. If the data is missing, I am not sure that you should include all the patients in these analyses.

Author comments: We acknowledge that there were issues with data completeness in this dataset. All staff received training on how to fill in the mHealth app. Therefore, attendees without recorded symptoms did not have the specified symptoms requested. However, we would also point out the difficulty of implementing a robust monitoring and evaluation platform in a resource-limited setting such as this, which is what the mHealth app was designed to address. We did not employ research staff but utilized the existing staff working in PHCs and QECH, who were trained on use of mHealth but were collecting data on top of their usual duties. We recognize the limitations of this dataset, but also believe that the comprehensive training and involvement of multiple stakeholders has led to a sizeable dataset that still provides insights into child acute illness seeking at the PHC level in a least developed country setting such as Malawi.

What is the difference between PHC mHealth triage and PHC clinician triage? Were they used for the same or different purposes? Were they done at (more or less) the same time? Why do they yield different outcomes? This needs to be explained.

Author comments: The healthcare system in Malawi is such that PHC care is provided by both clinically-trained and non-clinically trained health workers. PHC mHealth triage were data collected by non-clinically trained health workers (e.g. health surveillance assistants, security guards, other PHC staff), whereas PHC clinician triage were data collected by clinically trained health workers (clinical officers, nurses, medical assistants). As part of the same PHC visit, data were first collected by the non-clinical health worker using the mHealth application (mHealth triage), and then separately by the clinical health worker (clinician triage). We have clarified this in the methods (p5).

Abstract Methods: Needs rewriting. In its current form it does not give the information needed to understand the how the study was done.

Author comments: We have edited the abstract methods to clarify how the study was done (p2).

Abstract Results: The meaning of the term “successful referral” needs an explanation.

Author comments: We have edited the section to explain this (p2).

In table 1, PHC outcome, the sum of total referred from PHC does not equal the number referred after triage + referred after short stay. This needs an explanation or a correction.

Author comments: We have reviewed the variables and classifications we used for this and corrected this row (Table 1).

Discussion, p11 line 17-19: the sensitivity and specificity of the IMCI criteria: To what gold standard were they related?

Author comments: This sentence refers to the study by the WHO, which summarized the results of several studies evaluating IMCI guidelines and training of health workers in first-level health facilities. The standard for determining sensitivity and specificity in various studies were based on trainer assessment (Simoes 1997 Bull WHO 1997), diagnosis by the physician (Perkins 1997 Bull WHO), the paediatrician’s diagnosis (Weber 1997 Bull WHO), and paediatrician assessment (Kalter 1997 Bull WHO). It is worth mentioning that the IMCI publication considered “a key consideration in developing the IMCI guidelines was that they should be safe” – therefore, “to achieve adequate sensitivity in detecting severely ill children who require referral, criteria must be used that inevitably lead to some children being referred unnecessarily” (WHO 1997 Bull WHO).

Discussion, p11 line 42-45: These are not the same numbers as in table 1/figure 1.

Author comments: These data are listed in Table 2, under triage outcome (p21).

Discussion, p11 line 52-57: “case management guidelines are often not in line with national or international guidelines, which may explain our findings.” How does this explain your findings. Please elaborate.

Author comments: This sentence refers to a study in Kenya, where the case management guidelines that health workers followed did not align with national or international guidelines. The implication here is that health workers in this setting may not follow guidelines such as ETAT and IMCI, and

therefore accounting for the lower than expected referral rates of triaged Emergency cases (<60%), for example. We have clarified this in the text (p12).

Discussion, final paragraph: In your concluding remarks you state that “Our study provides quantitative data that allows us to identify gaps in care, which can form the basis to strengthen the referral pathway and improve child health outcomes.” Please be more concrete: Which gaps in care? How would you suggest to go about to strengthen the referral pathway?

Author comments: A critical component of the IMCI strategy is referral of severely ill children. Our findings that <60% of triaged Emergency cases were referred indicates a gap in healthcare delivery and would suggest that there are opportunities to strengthen the IMCI, ETAT, and referral training of healthcare workers. We had stated this in the discussion (p12) and the final paragraph was referring back to this comment. We have edited the final paragraph to make this more concrete (p13).

Supplementary tables 1 and 2 were missing.

Author comments: Supplementary Tables 1 and 2 are now included.

Reviewer: 2

Dr. Marlies van Houten, Spaarne Gasthuis

Comments to the Author:

good things:

- great way to track patients from remote PHCs to hospital in a low resource setting
- Great to quantify self-referral percentage
- great to see percentage of participants at each triage level referred or not

Author comments: Thank you, we agree that this study provides unique and important data on this vulnerable population.

Reviewer: 3

Dr. Andrew O'Regan, University of Limerick Faculty of Education and Health Sciences

Comments to the Author:

Dear authors,

I wish to commend you on an interesting, important and well conducted study. There are a number of points that need to be clarified before it is ready for publication in the BMJ Open. I have included these before and am confident that when they have been incorporated that you will have a strong, readable paper that will be of interest to a wide audience, and will make an impact.

While this is an excellent study and should be published, there are a number of important clarifications and revisions required to make the current draft more readable for a wider audience.

The title does not flow as well as it could- consider a slight rewording, such as 'paediatric presentations and referrals at a primary health centre in Malawi – a secondary analysis of ASPIRE'

Author comments: We have revised the title (p1).

Abstract

It is not clear to me from the abstract what the context and exact purpose of the study is. There is a 300 word count limit and consider expanding the abstract to explain what is meant by a successful referral. The results subsection of the abstract is confusing. If the introduction is re-written to state 'the aim of the study is to investigate the most common presentations, diagnoses and reasons for referral', then ensure that the Results section corresponds to the stated aim.

Author comments: We have revised the abstract, including what is meant by a successful referral, and the results section to more clearly correspond to the stated aim (p2).

What this study adds

Lines 24-31: does the 'referral rate' mean that of all the presentations to the PHCs that <2% were referred on to hospital? If yes, it should be explicitly stated. Does the line "...were sent, home and then self-referred to the referral centre" mean that they were discharged from the PHC and later attended a hospital? The wording is not clear. The term successful referral is explained later in the introduction but it would not be understood as it stands in the abstract.

Author comments: We revised this section to 'Strengths and limitations of this study' (p3). We have explained successful referral in this section, as well as in the abstract (p3).

Introduction

P 4 Line 6 – "under – 5 deaths" – rephrase please to – "mortality/death in children under five years"

P 4 line 56 – similarly, rephrase the term 'under-five children'

Author comments: We have edited this (p4).

Overall, I think the introduction could be enhanced by more reference to how children are assessed in community and in-hospital settings in SubSaharan Africa. As it stands, a reader unfamiliar with the subject would not have a strong enough context from the introduction. The authors identify the evidence gap in the literature and make the case for the study but the aim should be clearly stated and it would help if a sentence outlining the aim was followed by three bullet points that articulate clearly what the specific objectives are.

Author comments: We have edited the introduction to more clearly state the aim and objectives, and provided more references on paediatric emergency care in resource-limited settings (p4-5).

Methods

P 5 ETAT is used for the first time in the main manuscript – the full term should be used followed by ETAT in brackets (this has been done in the abstract but needs to be done again in the main manuscript).

Author comments: We have spelt out ETAT in first mention in the text (p5).

The terminology is confusing for the reader who is not familiar with SubSaharan Africa health care systems. The authors could clarify what is a primary health centre – how is it different to what is called a primary health care centre in Ireland or Britain? Are they specifically catered for children under five, or all children or everyone, are they walk-in, appointment-only, free, and who runs them doctors or trained physician assistants or nurses?

Author comments: We provide more information on the setting and staffing of primary health facilities in this setting (p5).

Similarly, is the term referral facility the standard term? Would referral hospital be more self-explanatory and do they mean specific paediatric units or general A and E?

Author comments: Referral facility refers to the next level of care facility, which is usually a secondary-level district hospital in Malawi. In urban Blantyre, QECH functions as the referral facility even though it is a tertiary-level hospital.

Results

From the introduction section, it appears that the study is concerned with children under five years but the results section appears to include the 41,876 children aged over five years. This needs to be clarified.

Author comments: We have clarified that the objective is to evaluate all children attending PHCs (p4).

Discussion

P9 line 42 – type ‘o’ -remove one ‘is’.

Phraseology needs to be improved in several places to ensure accuracy of what is being stated, e.g., ‘This study is the first time that successful referrals have been quantified in a systematic way along the referral pathway’... consider alternatively ‘this study is the first to report successful referrals that have been quantified in a systematic way’. That is phow I understand it.

Author comments: We have edited this (p10).

The limitations section is very good but the following paragraph should be rewritten as a conclusion so it should state exactly what the study contributed, i.e. rather than stating a fact that is already known “an effective referral pathway, to promote timely access, judicious allocation of human and physical resources, cost-effective care, and to optimise clinical outcomes”, that you have identified a gap in the pathway or rather a way of quantifying it and that this will contribute to a stronger, more seamless referral pathway.

Author comments: We have edited the conclusion (p13).

We thank the reviewers for their guidance and constructive feedback, and believe the revised manuscript has benefited greatly from the reviewer and editorial comments. We appreciate how constructive the editorial process has been.

VERSION 2 – REVIEW

REVIEWER	Vallersnes, Odd Martin University of Oslo, Department of General Practice
REVIEW RETURNED	28-Dec-2023

GENERAL COMMENTS	Though the manuscript is much improved, you still need to address the question of lack of data completeness more thoroughly in your Limitations section. In your response letter you explain the highly understandable reasons for this problem. However, in the Limitations section you need to describe how this affects the interpretation of your results. Very few patients are registered with a symptom at the PHC or the QECH (table 1), or a diagnosis at the QECH (tables 1 & 3). What about all the other patients? Did they not have symptoms? Or were they not diagnosed? Or are the data missing? And how do
---

	you interpret that? This needs to be stated and addressed in the Limitations section.
--	---

VERSION 2 – AUTHOR RESPONSE

Reviewer: 1

Dr. Odd Martin Vallersnes, University of Oslo, Oslo Accident and Emergency Outpatient Clinic

Comments to the Author:

Dear authors

Though the manuscript is much improved, you still need to address the question of lack of data completeness more thoroughly in your Limitations section. In your response letter you explain the highly understandable reasons for this problem. However, in the Limitations section you need to describe how this affects the interpretation of your results.

Author comments: We have added a sentence in the limitations section describing how this affects the interpretation of our results (p12-13).

Very few patients are registered with a symptom at the PHC or the QECH (table 1), or a diagnosis at the QECH (tables 1 & 3). What about all the other patients? Did they not have symptoms? Or were they not diagnosed? Or are the data missing? And how do you interpret that? This needs to be stated and addressed in the Limitations section.

Author comments: This refers to the issue of lack of data incompleteness as discussed above. Existing staff were trained in data collection using the mHealth app, but given the pragmatic nature of this study we do not have further information to account for the missing data. We have added a sentence in the limitations section describing how this affects the interpretation of our results (p12-13).